# Defining and Addressing Research Priorities in Cancer Cachexia through Transdisciplinary Collaboration

**DOI:** 10.3390/cancers16132364

**Published:** 2024-06-27

**Authors:** Margaret A. Park, Christopher J. Whelan, Sabeen Ahmed, Tabitha Boeringer, Joel Brown, Tiffany L. Carson, Sylvia L. Crowder, Kenneth Gage, Christopher Gregg, Daniel K. Jeong, Heather S. L. Jim, Andrew R. Judge, Tina M. Mason, Nathan Parker, Smitha Pillai, Aliya Qayyum, Sahana Rajasekhara, Ghulam Rasool, Sara M. Tinsley, Matthew B. Schabath, Paul Stewart, Jeffrey West, Patricia McDonald, Jennifer B. Permuth

**Affiliations:** 1Department of Gastrointestinal Oncology, H. Lee Moffitt Cancer Center & Research Institute, Tampa, FL 33612, USA; margaret.park@moffitt.org; 2Department of Biostatistics and Bioinformatics, H. Lee Moffitt Cancer Center & Research Institute, Tampa, FL 33612, USA; paul.stewart@moffitt.org; 3Department of Metabolism and Cancer Physiology, H. Lee Moffitt Cancer Center & Research Institute, Tampa, FL 33612, USA; virens@darwiniandynamics.org; 4Department of Machine Learning, H. Lee Moffitt Cancer Center & Research Institute, Tampa, FL 33612, USA; sabeen.ahmed@moffitt.org (S.A.); ghulam.rasool@moffitt.org (G.R.); 5Department of Drug Discovery, H. Lee Moffitt Cancer Center & Research Institute, Tampa, FL 33612, USA; tabitha.boeringer@moffitt.org (T.B.); smitha.ravindranadhanpillai@moffitt.org (S.P.); 6Department of Cancer Biology and Evolution, H. Lee Moffitt Cancer Center & Research Institute, Tampa, FL 33612, USA; joel.brown@moffitt.org (J.B.); jeffrey.west@moffitt.org (J.W.); 7Department of Integrated Mathematical Oncology, H. Lee Moffitt Cancer Center & Research Institute, Tampa, FL 33612, USA; 8Department of Health Outcomes and Behavior, H. Lee Moffitt Cancer Center & Research Institute, Tampa, FL 33612, USA; tiffany.carson@moffitt.org (T.L.C.); sylvia.crowder@moffitt.org (S.L.C.); heather.jim@moffitt.org (H.S.L.J.); nathan.parker@moffitt.org (N.P.); sara.tinsleyvance@moffitt.org (S.M.T.); 9Department of Diagnostic Imaging and Interventional Radiology, H. Lee Moffitt Cancer Center & Research Institute, Tampa, FL 33612, USA; kenneth.gage@moffitt.org (K.G.); daniel.jeong@moffitt.org (D.K.J.); aliya.qayyum@moffitt.org (A.Q.); 10School of Medicine, University of Utah, Salt Lake City, UT 84113, USA; chris@storylinehealth.com; 11Department of Physical Therapy, University of Florida, Gainesville, FL 32610, USA; arjudge@phhp.ufl.edu; 12Department of Nursing Research, H. Lee Moffitt Cancer Center & Research Institute, Tampa, FL 33612, USA; tina.mason@moffitt.org; 13Department of Supportive Care Medicine, H. Lee Moffitt Cancer Center & Research Institute, Tampa, FL 33612, USA; sahana.rajasekhara@moffitt.org; 14Department of Malignant Hematology, H. Lee Moffitt Cancer Center & Research Institute, Tampa, FL 33612, USA; 15Department of Cancer Epidemiology, H. Lee Moffitt Cancer Center & Research Institute, Tampa, FL 33612, USA; matthew.schabath@moffitt.org; 16Lexicon Pharmaceuticals, Inc., Woodlands, TX 77381, USA

**Keywords:** cancer-associated cachexia, body composition, biomarkers, exercise, nutrition, patient-reported outcomes, supportive care, machine learning, mathematical modeling

## Abstract

**Simple Summary:**

Cachexia occurs in up to 80% of patients with cancer. Although cancer-associated cachexia dramatically decreases overall survival and quality of life, it is often overlooked. To make meaningful progress in identifying cancer cachexia earlier and finding treatments for this condition, Moffitt Cancer Center held its first Cachexia Working Group Retreat in 2022. This manuscript describes the priorities discussed at the retreat and highlights collaborations that arose afterward.

**Abstract:**

For many patients, the cancer continuum includes a syndrome known as cancer-associated cachexia (CAC), which encompasses the unintended loss of body weight and muscle mass, and is often associated with fat loss, decreased appetite, lower tolerance and poorer response to treatment, poor quality of life, and reduced survival. Unfortunately, there are no effective therapeutic interventions to completely reverse cancer cachexia and no FDA-approved pharmacologic agents; hence, new approaches are urgently needed. In May of 2022, researchers and clinicians from Moffitt Cancer Center held an inaugural retreat on CAC that aimed to review the state of the science, identify knowledge gaps and research priorities, and foster transdisciplinary collaborative research projects. This review summarizes research priorities that emerged from the retreat, examples of ongoing collaborations, and opportunities to move science forward. The highest priorities identified include the need to (1) evaluate patient-reported outcome (PRO) measures obtained in clinical practice and assess their use in improving CAC-related outcomes; (2) identify biomarkers (imaging, molecular, and/or behavioral) and novel analytic approaches to accurately predict the early onset of CAC and its progression; and (3) develop and test interventions (pharmacologic, nutritional, exercise-based, and through mathematical modeling) to prevent CAC progression and improve associated symptoms and outcomes.

## 1. Introduction

Cancer patients commonly experience fatigue, loss of appetite, and dramatic weight loss owing to the wasting of muscle and fat tissue. This ‘wasting’ syndrome, known as ‘cachexia’, greatly decreases quality of life (QoL) among patients with cancer and negatively impacts their psychological and physical health, leading to poorer response to treatment and reduced survival [1]. It is estimated that cachexia affects 50–80% of all cancer patients and accounts for up to 30% of cancer deaths [2]. Compelling evidence suggests that cancer-associated cachexia (CAC) is a multifactorial metabolic syndrome, requiring multimodal therapy targeting different factors involved in its pathophysiology to improve QoL, tolerance and response to therapy, and increased overall survival [3]. Despite an international consensus definition of cachexia [3] wherein CAC is described as a spectrum consisting of three clinically relevant stages: (i) pre-cachexia (metabolic changes or anorexia without significant weight loss); (ii) cachexia (weight loss >5% in the past 6 months—excluding simple starvation or body mass index (BMI) <20 and any degree of weight loss >2% or skeletal muscle index consistent with sarcopenia and any degree of weight loss >2%); and (iii) refractory cachexia (any measure of cachectic weight loss described above concomitant with a low performance score), and the completion of several large phase III intervention trials, there is currently only one FDA-approved treatment for CAC in the United States; thus, CAC remains an urgent unmet medical need and an important clinical challenge in cancer therapy [4,5,6].

In alignment with management recommendations for CC from the American Society of Clinical Oncology (ASCO) [4], which calls for the need to address the lack of effective methods for the early diagnosis and treatment of CAC to improve patient outcomes, in May 2021, investigators from our cancer center (H. Lee Moffitt Cancer Center and Research Institute in Tampa, Florida) founded the ‘Moffitt Cancer Cachexia Initiative (MCCI)’. This initiative brings together a diverse team of clinical, basic, population, and quantitative scientists with expertise in oncology, radiology, physiology, exercise, nutrition, epidemiology, computer science, palliative care, and basic and translational science with external collaborators with extensive expertise in cachexia research for actions aimed at prevention, early detection, or reversal of cachexia using a multimodal approach.

Working groups have assembled as part of other initiatives to prioritize key CAC research areas. For example, the LUNGevity Foundation [7], University of Rochester [8], and Sharing Progress in Cancer Care [7] working groups identified priority areas including better defining and measuring cachexia; refining knowledge around CAC clinical trials and selecting appropriate endpoints to enable the demonstration of effective therapies; expanding education about CAC among patients, caregivers, and clinicians; and increasing evidence, resources, and insurance coverage for nutritional support and physical therapy for patients with cancer cachexia. By implementing feedback that had been previously published by two of these working groups [8,9], in May 2022, our MCCI executive committee led an in-person inaugural working group retreat on cancer cachexia at Moffitt (Tampa, Florida) to brainstorm opportunities to foster collaborations across disciplines, with goals of reviewing the state of the science, identifying knowledge gaps that build upon and/or complement those identified by others [8,9], and formulating investigator-led research projects through active discussion and consensus. To support these efforts, three external collaborators with extensive expertise in cachexia and behavioral oncology were also invited to present and participate in the retreat (Dr. Teresa A. Zimmers, Oregon Health and Sciences University; Dr. Andrew Judge, University of Florida, and Dr. Chris Gregg, University of Utah).

### Key Research Priorities and Opportunities

Three research priorities emerged from the retreat and are summarized in our graphical abstract: (1) Evaluation of patient-reported outcome (PRO) measures and examination of their use in predicting cancer cachexia-related outcomes. (2) Identification of novel classes of biomarkers and analytic approaches that can be used to accurately predict the early onset of cachexia and/or its progression; (3) Development and testing of multimodal interventions to reduce symptom burden of cachexia and improve QoL, tolerance to therapy, and increase overall survival.

In the sections that follow, we summarize these priorities and related topics covered at our retreat, highlight examples of past and ongoing pilot projects and initiatives, and summarize ideas that developed organically through discussions and new transdisciplinary collaborations.

## 2. Research Priority 1: Evaluation of PROs as Predictors of Cachexia Status and Clinical Outcomes

### 2.1. Overview of PRO and Supportive Care Needs

PROs are data reported directly by patients [10]. PRO measures allow for reporting of complex biological and clinical symptoms such as pain, dyspnea, fatigue, and nausea/vomiting, with opportunities for patients to document these concerns routinely via online surveys from the comfort of their own home or in clinic [11]. Touch-screen tablets and internet-based data collection techniques have increased the feasibility of collecting, storing, analyzing, and reporting PROs in real time in clinical practice.

As no curative drugs are available yet and current treatments are only palliatives, emphasis is often placed on PROs that can facilitate reporting and palliation of symptoms and improvement or maintenance of QoL, encompassing physical, emotional, cognitive, and social components [11]. Opportunities to assess the value of PROs in CAC research include the identification of longitudinal associations of changes in PROs with clinical outcomes, developing risk assessment algorithms to determine patients at risk for CAC, and using PRO measures in CAC randomized clinical trials.

Supportive cancer care is defined as ‘the prevention and management of the adverse effects of cancer and its treatment’ [12] and constitutes a key component of oncological care. This discipline aims to maintain or enhance QoL from diagnosis through treatment to post-treatment care with improvement in QoL and life prolongation as its main goals [13]. The challenges in CAC, including its complexity, the absence of clear diagnostic criteria, and the lack of robust evidence guiding clinical practice, hinder reliable diagnosis and management. Additionally, the absence of a validated screening tool and consistent malnutrition screening, the lack of a collection of PROs, along with knowledge gaps among clinicians, often result in the identification of cachexia only in advanced stages [8]. Due to a lack of robust evidence, no specific pharmacological agent was endorsed by ASCO in their 2020 published guidelines [4]. Although dietary counseling with or without nutritional supplementation was cited as helpful, it was not shown to significantly slow or reverse disease [4]. However, a recent ASCO guideline rapid update from an expert panel [5] recommends that clinicians offer low-dose olanzapine (an FDA-approved anti-psychotic known to promote weight gain) once daily to stimulate appetite and improve weight gain for patients with advanced cancer. For patients who cannot tolerate low-dose olanzapine, a short-term trial of a progesterone analog or a corticosteroid is recommended [5]. This therapeutic approach, however, is also limited owing to secondary toxicities, including muscle atrophy.

Given that supportive care specialists have expertise in symptom control and psychosocial support, introducing their expertise early in oncology care is crucial [13]. To improve cancer care delivery, Moffitt Cancer Center’s Supportive Care Service in collaboration with Moffitt’s gastrointestinal (GI) cancer service has implemented an integrated supportive care service model within the gastrointestinal (GI) cancer clinic, focused on providing early, efficient, and comprehensive supportive care for GI cancer patients. This supportive care service model facilitates early assessment for malnutrition and cachexia while addressing other secondary nutritional impact symptoms such as alterations in taste and smell (due to, e.g., chemotherapy), constipation, and bowel pain [14]. Moffitt intends to build upon this supportive care service model to improve the overall care of patients with cachexia.

### 2.2. PROs in Nutritional Interventions for Cancer Cachexia

Validated PROs allow for the measurement of dietary intake and include instruments such as the patient-generated subjective global assessment short form (PG-SGA-SF) [15] and Ingesta Score [16]. These questionnaires unify measurements of food intake, which can inform clinical practice to best identify patients at risk for malnutrition and cachexia. Early and routine screening allows optimization of symptom management and nutrition. The PG-SGA-SF and Ingesta scores can also identify patient characteristics most associated with weight loss, appetite loss, and reduced food intake and can be prospectively validated [11]. For instance, appetite loss is thought to be central in the CAC pathophysiology [17]. CAC is hypothesized to impact the hypothalamus in a way that decreases feelings of hunger and the desire to eat [18]. Alterations of this biological mechanism can then lower overall food intake and contribute to accelerated weight loss. Through early and routine screening, independent PRO predictors of CAC can help clinicians identify patients who may need to be followed more closely to ensure early and appropriate therapeutic intervention.

### 2.3. PROs as Essential in Identifying the Cancer Cachexia Stage and Predicting Various Endpoints in a Population-Based Study of Pancreatic Cancer

The prevalence of CAC is highest in GI malignancies and pancreatic ductal adenocarcinoma (PDAC) in particular, with up to 80% of PDAC patients affected during the course of their disease [19]. Despite CAC being a continuum that spans from pre-cachexia, cachexia, and refractory disease, most existing studies of PDAC-associated cachexia have classified cachexia stage as cachectic or non-cachectic [20,21,22,23,24,25,26], and most do not evaluate PRO as endpoints as recommended by ASCO [4]. Vigano and colleagues [27] developed a CAC classification system that considers patient-reported data on changes in weight, appetite, food intake, activity level, and physical function along with laboratory values, including C-reactive protein, white blood cell counts, albumin, and hemoglobin levels. Our team leveraged this classification system [27] and data obtained at the point of care to determine the prevalence of the CAC stage at diagnosis and PROs that document symptom burden, supportive care needs, and QoL, among a diverse cohort of incident PDAC cases. Patients with cancer were recruited to the Florida Pancreas Collaborative (FPC), a multi-institutional population-based prospective cohort study and biobank of patients of Non-Hispanic Black (NHB), Non-Hispanic White (NHW), or Hispanic/Latinx (H/L) background [28]. FPC data collection occurred at baseline/enrollment and at 6- and 12-months post-baseline and included instruments such as the abridged version of the Patient-Generated Subjective Global Assessment (aPG-SGA) [29], a revised version of the Edmonton Symptom Assessment System (ESAS-r) [30], and the Canadian Problem Checklist [31,32] to allow for self-reported assessment of height, weight, changes in weight, food intake, presence of symptoms affecting food intake, activities and function, and supportive care needs. QoL was assessed using the European Organization for Research and Treatment of Cancer (EORTC) QLQ-C30 and QLQ-PAN-26 instruments [33,34,35]. Among the 309 patients in the analysis, the overall prevalence of pre-cachexia, cachexia, and refractory cachexia was 24.6%, 54.1%, and 8.4%, respectively [36]. The estimate of pre-cachexia is higher than that reported in other studies of PDAC that considered only weight loss [37]. As expected, the symptom burden increased as the cachexia stage increased. The most frequently reported symptoms at the time of PDAC diagnosis across racial/ethnic groups and sex included weight loss, fatigue, pain, anxiety, and depression, with pain significantly worsening over time. Collectively, findings from the FPC suggest a significant number of individuals with pre-cachexia could be targeted for early intervention with a multidisciplinary approach that includes symptom management, nutrition, physical therapy, psychosocial support, resistance training, and targeted therapeutic agents [36].

### 2.4. PRO and Radiomics

Changes in PROs also aid in predicting how cancer patients will respond to treatment. We previously evaluated whether PRO dynamics could be used as inter-radiographic predictors of tumor volume changes among non-small cell lung cancer (NSCLC) patients treated with immunotherapy (IO) [38]. Prior studies demonstrated that temporal changes in PROs can be leading indicators of clinically important events such as cancer development, treatment progression, and survival [39,40,41,42,43,44,45]. Moreover, a phase III RCT that compared weekly web-based self-scored patient symptoms versus routine surveillance among patients with advanced lung cancer found cancer progression was detected 5 weeks earlier and overall survival (OS) was extended by 9 months among patients in the PRO group [46]. Building on this emerging area of research, in this collaborative project, it was hypothesized that change in PROs could be used as an early predictor of progressive disease. To test this hypothesis, PRO dynamics and CT-derived tumor volume changes were analyzed among 85 patients with NSCLC undergoing IO [38]. PRO questionnaires and tumor volume scans were completed biweekly and monthly, respectively. Correlation and predictive analysis were conducted to identify specific PROs that could accurately predict patient response. Changes in tumor volume over time were statistically significantly correlated with dizziness (*p* < 0.005), insomnia (*p* < 0.05), and fatigue (*p* < 0.05). Cumulative changes in insomnia predicted, on average, progressive disease with 77% accuracy 45 days prior to the next imaging scan. This published pilot study [38] presents the first time that patient-specific PRO dynamics have been considered to predict how individual patients will respond to treatment. A multi-institutional study involving a larger, more diverse patient population is planned to validate these findings and to include the collection of longitudinal weights since prior studies have demonstrated this as a reliable indicator of patient outcomes [47,48].

## 3. Research Priority 2: Identification of Novel Classes of Biomarkers and Analytic Approaches That Can Be Used to Accurately Predict the Early Onset of Cachexia and Its Progression

Biomarkers are quantifiable indicators of a normal biological or physiopathological state and in the clinical setting may be used for diagnosis, prognosis, and monitoring of disease progression, thereby serving as predictors of future status, clinically relevant disease midpoints or endpoints, and pharmacologic and/or psychological responses [49,50,51]. Retreat participants discussed the importance of evaluating various classes of biomarkers (molecular, imaging, behavioral) using approaches that take advantage of ‘omics’ technologies, machine learning (ML)/artificial intelligence (AI) algorithms, and mathematical modeling.

### 3.1. Molecular Biomarkers and ‘Omics’ Analyses

Molecular biomarkers include proteins, amino acids, fatty acids nucleic acids, and other molecules that may be harvested from tissue or liquid biopsies such as blood, saliva, urine, or stool to measure various clinical phenotypes [52]. In CAC, there is a dire need for early biomarkers of CAC. Although there are currently no universally accepted and approved biomarkers of any kind for CAC [53,54], recent studies have made significant progress in identifying metabolic differences between cancer-induced versus chemotherapy-induced cachexia [55], and shifts in metabolic signatures that appear before observable weight loss [56].

Broadly speaking, ‘-omics’ approaches cover many areas of research that include genomics, transcriptomics, proteomics, metabolomics, lipidomics, and epigenomics, allowing for global analyses of genes, RNA, proteins, metabolites, lipids, and chemical modifications of DNA or histone proteins, respectively. In recent years, the use of ‘-omics’ technologies in the analysis of clinical samples of numerous disease states has grown significantly, and CAC is no exception. One defining factor for cancer-related cachexia is that, in addition to the tumor/tumor microenvironment and circulation, multiple tissue types are involved in CAC progression, providing motivation to carry out omics-based analysis on tissues that influence or are influenced by cachexia such as muscle and adipose. In this regard, the laboratory of Dr. Zimmers recently carried out transcriptomics analysis of muscle and adipose tissue biopsies (taken from the same individuals) of PDAC patients [57]. This study identified differentially expressed genes (DEGs) in the muscle and adipose tissues of cachectic PDAC patients, convergent pathways in age-related muscle wasting, and muscle wasting secondary to CAC [57]. Furthermore, their data suggested that, although the set of DEGs in muscle tissue largely differed from those in adipose tissue from cachectic patients, many dysregulated pathways (including senescence pathways) were common to both tissues [57]. Finally, in preclinical studies, the Zimmers’ lab found that a larger number of genes were dysregulated in cachexia-related pathways at early-stage in male mice compared to female mice, suggesting sex-specific mechanisms are at play and are influenced by the activity of the reproductive hormone and TGFβ family member Activin [58].

Ebhardt et al. [59] took a proteomics approach to interrogate muscle biopsies from individuals with age-related sarcopenia, healthy non-sarcopenic individuals, weight-stable GI cancer patients, and GI cancer patients with CAC. Comparing the proteomes of these patient groups led to the identification of changes in the expression of clusters of proteins specific to CAC including those involved in ATP production and myosin regulation, providing a protein signature that was unique to patients with cachexia [59]. As a wasting disease, CAC involves many complex metabolic perturbations. In light of the numerous metabolic perturbations, Miller et al. [60] pursued a metabolomics approach and observed increases in plasma lysolipids such as medium-chain lysophosphatidylcholine (LPC) glycerophospholipids in weight-loss versus weight-stable patients. Similarly, Cala et al. [61] identified significant decreases in short-chain LPC as well as decreases in amino acids, specifically derivatives of histidine, lysine, and tryptophan that also appear to be associated with cachexia [61]. We believe the application of multi-omics will be especially powerful when combined with quantitative body composition data from various imaging modalities as described next.

### 3.2. Imaging Biomarkers and the Application of AI/ML Tools

Imaging modalities also hold promise for diagnosing and monitoring cancer cachexia through the analysis of changes in body and tissue composition [62]. As summarized by others [62,63], computed tomography, magnetic resonance imaging, fluoro-2-deoxy-D-glucose (^18^FDG) PET, and dual-energy X-ray absorptiometry (DEXA) have utility in the setting of cancer cachexia, with each having their advantages and limitations. Quantitative and qualitative changes to adipose and muscle tissue and other organs also present important biomarkers for phenotyping cachexia and determining its onset and progression, especially with the integration of novel machine learning (ML) approaches that can learn sub-visual patterns (or equivalent representations) from the annotated data and use these learned representations to make more accurate decisions/predictions [64]. While some conventional machine learning and statistical approaches can operate with minimal or no training datasets, respectively, they often require significant manual effort for image curation and processing, feature selection, and engineering. Examples of such manual feature engineering include manually setting the intensity thresholds for pixels, usually in Hounsfield Units, to identify particular structures in the CT images. Some initial research studies used threshold-based methods to identify adipose tissue [65,66,67], whereas atlas-based techniques were proposed to identify abdominal muscles [68]. However, these manual feature engineering approaches pose challenges. They are time-consuming, subjective, and often not reproducible. In contrast, modern ML methods are data-driven and commonly called deep learning (DL), owing to a large number of hierarchical data processing layers. These models are designed to automatically learn distinct data features based on a suitable supervisory signal (e.g., annotations). DL models have a large parameter space (in millions) and thus possess a superior capacity to learn nuanced, subtle, sub-visual features that are hard to hand-engineer or encode manually. Various DL models, including convolutional neural networks (CNNs), have been proposed to process CT images directly without the need to calculate hand-engineered features (e.g., shape, size, texture, etc.). CNNs and their variants are used to accurately identify skeletal muscles and adipose tissue at the pixel level [69,70,71,72,73]. A CNN-based architecture called the UNet is a popular architecture used for segmentation tasks in medical imaging owing to its effectiveness and superior performance.

A collaborative pilot project aims to develop an automated pipeline based on the DL model to identify muscle and various adipose tissues (visceral, intra-muscular, and subcutaneous) from abdominal CT scans of patients with gastroesophageal cancers. The team developed a “dataset” of CT images with pixel-level annotations for mutually exclusive labels (e.g., muscle, visceral fat, intra-muscular fat, subcutaneous fat, or background). The team used SliceOmatic software (Tomovision, Montreal, QC, Canada) with HU thresholding to generate segmentations/annotations that are manually verified and corrected (if required) by a radiologist. A single data point in the dataset consists of an axial CT image belonging to a study of one patient and corresponds to the middle of the L3 vertebral segment of the axial CT series. An abdominal radiologist visually identified the axial mid-L3 CT slice used for annotation. The dataset consists of serial images for most patients to provide a longitudinal view of the changes in the skeletal muscles and fat. A total of 45 annotated images have been used to train the deep learning model using five-fold cross-validation.

The trained model can segment muscle and three types of fat in a fraction of a second with an average accuracy of 96% for the skeletal muscle, 98% for the subcutaneous fat, 93% for the visceral fat, and 81% for the intra-muscular fat for the test dataset, which consists of 13 CT images. The segmentation results were further used to calculate (1) the skeletal muscle index (SMI), (2) the ratio of visceral to subcutaneous fat, and (3) the ratio of visceral to total fat.

The ML/AI team is working closely with Moffitt Health Data Services and Flywheel teams to build an automated pipeline using Python scripts called “gears” in the Flywheel Enterprise platform as shown in Figure 1a. The process of importing data from the PACS/VNA into Flywheel Discovery is a manual process managed by data brokers. The gears run sequentially on the imported data to perform various tasks, including (1) identify the required axial series (such as intravenous contrast phase), (2) preprocess data, (3) tag slices corresponding to each lumbar level, and (4) predict skeletal muscle segmentation corresponding to each tagged slice using a trained deep learning model, which also outputs an uncertainty map depicting the model confidence scores at the pixel level. In collaboration with other investigators from the MCCI working group, the ML/AI team is also evaluating the DL model on other cancers, including pancreatic, colorectal, and ovarian. Figure 1b shows the model output for three representative mid-L3 sample slices, one from each of the gastroesophageal, colorectal, and pancreatic cancers. This pipeline will allow researchers to use muscle and fat segmentation tools with minimal manual work and integrate the CT-based muscle and adipose tissue segmentation with the patient’s electronic medical records, enabling triaging and referral for nutrition and exercise interventions [74].

### 3.3. Behavioral Biomarkers

The group also discussed an online platform called Storyline Health (https://storylinehealth.com/ (accessed on 27 February 2024)). Based on research that operates at the intersection of neurobiology, genomics, and ML [75,76,77], this innovative online tool provides an ML or AI platform that allows patients to gain control over their own healthcare. It works by analyzing cognitive and behavioral patterns where patients answer questions about how they are feeling, how their disease experience has changed over time, what treatments they have received, and how their disease has responded to those treatments. Storyline measures over 20,000 different micro-features of speech, voice, and motor patterns from video assessments to uncover biomarkers of illness, drug responses, and predictive models of outcomes. Storyline is an app that can be accessed on a smartphone, a tablet, and many other devices and has been designed to enable efficient and intelligent features, including precision care pathways, messaging, and communications, with military-grade security for HIPAA, GDPR, and HITECH.

Through Storyline, patients and their physicians can tap into a growing library of clinical assessments, automated triggers and scoring, e-signatures and consents, and advanced behavioral AI. ML will aid researchers in identifying subgroups of patients with similar disease progressions and responses to treatment, which will help guide effective therapies in a manner tailored to each specific patient. Thus, Storyline enables massively scalable precision medicine and is well-suited to help solve the problem of detecting early-stage cachexia, monitoring symptoms of cachexia, and building predictive models of cachexia and cancer progression to help improve care decisions.

### 3.4. Leveraging Mathematical Modeling to Predict Cachexia Onset and Progression

In addition to efforts to interrogate molecular, imaging, and behavioral biomarkers through omics and ML approaches, our specialists in evolutionary biology and mathematical oncology, cancer biology, computer science, and informatics are working collaboratively to anticipate a tumor’s path and harness the body’s own immune system to fight cancer. Efforts to integrate these broad perspectives to address CAC have grown out of the 2022 Integrated Mathematical Oncology Workshop, in which a team of math modelers, working with basic researchers, oncologists, radiologists, and evolutionary ecologists, developed complementary mathematical models for the induction and progression of CAC.

Modeling efforts focus on capturing two hypotheses for what drives cancer cachexia: (1) liver-centric and (2) tumor-centric. The liver-centric model sees the progression of CAC as emerging when the combined metabolic demands of the patient and their tumor burden exceed the liver’s capacity to resupply nutrients to the blood and remove and process metabolites and toxins [78,79,80,81,82,83]. To meet this excess demand, the liver induces new supplies by signaling the breakdown of muscle and adipose tissue. In the absence of cancer, this can be an adaptive response of the body to any acute, excessive need in response to injuries of the lung, skin, or heart. But, with continued tumor growth, the excess demand on the liver increases over time driving the progressive loss of muscle and adipose tissue. In contrast, the tumor-centric model views cachexia as a consequence of an imbalance between pro-inflammatory and anti-inflammatory cytokines [82,84,85,86].

These two hypotheses can be captured in a single phenomenological model of the disruption of the body’s resource supply dynamics during tumor progression:
$$\dot{R}=\underset{digestion-demand}{\underbrace{u-d}}+\underset{resource\;replenishing}{\underbrace{q\left(R^{\prime }-R\right)}}-\underset{tumor\;drawdown}{\underbrace{aRT}}+\underset{muscle\;drawdown}{\underbrace{cM}}$$

where 
c=1
 if 
$q(R^{\prime }-R)>L$
 (cachectic) and 
c=0
 otherwise (non-cachectic).

Resource and nutrient availability, *R*, is replenished as a function of digestion, *u*, lessened by daily demand, *d*, plus a liver-controlled replenishment in proportion to the mismatch in current availability to desired homeostatic level, *R’*, scaled by a rate parameter *q*. In this model, the tumor draws down resource supply in proportion to tumor size and availability of total resources. Muscle drawdown happens when resource replenishment is rate-limited.

Cachexia is thus defined in the liver-centric model as the imbalance in resource replenishment [87,88]. Above a certain rate of required replenishment, liver function is rate-limited (
L
) and begins to draw down muscle. The liver-centric model hypothesizes that metabolic dysregulation resulting from tumor demand for resources and release of potentially toxic metabolites directly overloads the critical functions of the liver (Figure 2A). Consequently, the liver signals first muscle, and then adipose tissue for amino acids and then fatty acids, and this over-taxing of liver function directly causes the wasting seen in CAC.

Alternatively, this same phenomenological model can incorporate the tumor-centric hypothesis, where the tumor directly induces CAC via a cytokine storm that drives systemic inflammation [89], which in turn dysregulates muscle and adipose tissue homeostasis, resulting in chronic wasting (Figure 2B) [87]. This manifests as an increased drawdown rate by the tumor, increasing the cachectic rate as a function of induced systemic inflammation.

The next steps include analyzing more fully the model’s behavior with stability and parameter sensitivity properties to generate hypotheses that can be tested with preclinical animal models, with patient data, or ideally, with both animal models and patient data (manuscript in preparation). Ultimately, we believe this integrative and transdisciplinary approach can identify both underlying mechanisms and potential treatments to counter the development and progression of cancer cachexia.

### 3.5. Important Considerations for Cachexia Biomarker Research and Modeling

Retreat participants also emphasized the need for focused evaluation of (a) biomarkers in the context of hematologic malignancies and (b) sociodemographic characteristics that may influence health disparities.

#### 3.5.1. Biomarker Research Specific to Hematologic Malignancies

Little is known about cancer-related cachexia and sarcopenia in hematologic malignancies. One limitation of cancer-related cachexia in hematologic malignancy research is the use of different terminology for comparison between studies, with cachexia, sarcopenia, and adipopenia used interchangeably. Like solid tumors, cancer-related cachexia and sarcopenia are associated with decreased overall survival and progression-free survival, with most research being derived from aggressive lymphomas [90]. Decreased survival could be related to the decreased ability to tolerate treatment. Research efforts are focused on identifying patients who have cancer-related cachexia and sarcopenia or are pre-cachectic. Various indices have been developed to determine prognosis more accurately by including nutritional parameters in addition to traditional staging. These include a prognostic nutritional index [91], a cachexia index [92], muscle density evaluation [93], and the Glasgow Prognostic Index [94]. All these nutritional prognostic indices use CT scans for the determination of fat and muscle with varying cutoff levels at differing vertebral and muscle areas. A consistent finding independent of the index is that patients who are cachectic and/or sarcopenic have inferior outcomes.

Blood-based biomarkers have also been explored for their ability to improve understanding and identification of cancer-related cachexia among patients with hematologic malignancies. Some markers are associated with an inflammatory state and include C-reactive protein (CRP), hemoglobin, and albumin [95]. In one study [96], CRP was increased to a mean of 34.18 in cachectic patients and 17.57 (*p* = 0.021) in non-cachectic patients. Hemoglobin and albumin levels were significantly lower in the cachectic group, *p* = 0.004. In another study, >54 appeared to identify risk of cachexia among patients with various hematologic malignancies [96], whereas a third study reported CRP > 10 was significantly associated with cachexia (*p* = 0.020) [1]. Thus, CRP appears to have the potential to identify patients with cancer-related cachexia. However, as noted, the threshold for diagnostic CRP levels varies, and CRP levels are affected by other clinical conditions as well as lifestyle, making a definitive threshold challenging. This indicates the need to find additional blood-based biomarkers that will help distinguish CRP levels driven by (or associated with) CAC from other underlying causes.

In a single study focused on patients with myelofibrosis, a hematologic malignancy associated with an inflammatory state, cachexia predictive markers for survival were cholesterol at 122 mg/dL and an albumin level of 4.3 g/dL [97]. This has not been replicated in other hematologic malignancies. Taken together, a best-risk model for the identification of patients with hematologic malignancies at risk for cachexia has not been identified. Further research is needed to fill this knowledge gap and develop and implement interventions early to improve outcomes.

#### 3.5.2. Evaluation of Inequities Based on Sex, Race, Ethnicity, and Insurance Status

Despite the high prevalence and impact of CAC on clinical outcomes, limited data exist regarding possible health inequities in CAC development and progression. Focused evaluation of factors including biological sex, race, ethnicity, and socioeconomic status may lead to novel strategies for addressing cancer health inequities. *Sex.* As we and others have shown [20,98,99,100,101,102,103,104,105], male cancer patients generally have a higher CAC prevalence, greater weight loss or muscle wasting, differential response to treatments, and worse outcomes than female patients. These disparate outcomes can partly be attributed to the fact men generally have greater muscle mass and other sex-associated traits underlying physiological and pathological conditions [106,107]. Indeed, across cancer types, the data support sex-specific differences in CAC phenotypes involving skeletal muscle fiber type, function, and metabolism [57,58,106,108,109,110,111,112,113,114]. Understanding the basic mechanisms underlying these sex-based differences, also referred to as “sexual dimorphism”, remains a critical gap that needs to be addressed to inform tailored interventions and guidelines to prevent or mitigate the effects of cancer, CAC, and its treatment in a sex-specific manner. *Race and ethnicity.* We know only a few studies [98,104,115] that specifically sought to explore possible racial or ethnic differences in CAC incidence and/or survival outcomes. In a retrospective chart review from a prospective tumor registry of 882 patients from Texas with gastroesophageal or colorectal cancer diagnosed between 2006 and 2013, Olaechea et al. [104] found that non-Hispanic Black (NHB, odds ratio (OR), 2.447; *p* < 0.0001) and Hispanic/Latinx (H/L) (OR, 3.039; *p* < 0.0001) patients were at an approximately 150% and 200%, respectively, greater risk of presenting with cachexia than non-Hispanic White (NHW) patients after controlling for potential confounders including age, sex, comorbidities, tumor site, histology, and stage. In a small study of NHB compared to NHW matched on stage, our team [98] found that markers of pancreatic cancer-induced cachexia were more frequent and included greater decreases in core musculature compared to corresponding healthy control patients (25.0% vs. 10.1% lower), greater decreases in psoas musculature over time (10.5% vs. 4.8% loss), lower baseline serum albumin levels (3.8 vs. 4.0 gm/dL), and higher platelet counts (332.8 vs. 268.7 k/UL) [98]. In a retrospective analysis of 957 patients with stage IV non-small cell lung cancer (NSCLC) between 2014 and 2020 in Texas, Olaechea et al. [115] also showed that Black race and Hispanic ethnicity were independently associated with more than a 70% increased risk of presenting with cachexia at the time of NSCLC diagnosis (*p* < 0.05). NHB patients presented with stage IV disease around 3 years younger than NHW (Kruskal–Wallis *p* = 0.0012; *t* test *p* = 0.0002), with cachexia status at diagnosis significantly predicting poor survival [115]. *Insurance status.* Among patients with GI cancers [104] and NSCLC [115], the absence of private insurance coverage was also associated with elevated cachexia risk compared to privately insured patients, especially among Blacks [115].

Collectively, these findings suggest that these covariates may help identify individuals disproportionately affected by cachexia development and progression and associated outcomes. The possibility remains that differences in rates of CAC may be related to biological differences related to race, ethnicity, sex, and healthcare inequities. It is clear that additional studies are needed to evaluate the relationship between determinants of health inequities in CAC incidence and survival across minority and socioeconomically disadvantaged patients affected by solid and hematologic malignancies. It is recommended that multi-institutional, longitudinal studies be designed to collect and integrate data related to possible social and biological determinants of CAC. To mitigate cancer disparities, it also seems prudent to study and address factors highlighted by Olaechea et al. [104], which include financial burden, chronic stress, and transportation needs.

## 4. Research Priority 3: Development and Testing of Interventions (Pharmacologic, Nutritional, and Exercise-Based) to Reduce Symptoms and Side Effects of Cachexia and Improve QoL, Tolerance to Therapy, and Increase Overall Survival

From a drug discovery perspective, the aim has been to develop pharmacological agents that diminish inflammation, improve appetite, maintain/increase weight, and decrease muscle wasting in CAC patients ultimately improving QoL and OS. However, developing a single agent to combat this multifactorial disease has proved to be extremely challenging. Due to a lack of robust evidence of efficacy, ASCO did not endorse any specific pharmacological agent as the standard of care for CAC in their 2020 published guidelines [4]. ASCO did note, however, that dietary counseling with or without nutritional supplementation is helpful but limited as it was not shown to significantly slow or reverse disease [4]. Retreat participants reviewed current therapeutic approaches and recognized that while working to fill the unmet need for FDA-approved pharmaceutical agents [4,5], great merit exists for the continued development and testing of, (a) novel CAC animal model systems, (b) nonpharmaceutical interventions such as nutrition and exercise/physical activity, and (c) innovative strategies that leverage lessons from nature and provide a framework to understand and combat cachexia-related outcomes.

### 4.1. Overview of Pharmacologic Interventions

#### 4.1.1. Olanzapine

Although ASCO did not recommend any pharmacologic agent in their 2020 guidelines, a recent ASCO guideline rapid update (2023) [5] now recommends that clinicians offer once daily, low-dose olanzapine to patients with advanced cancer. Olanzapine, an FDA-approved atypical anti-psychotic known to promote significant weight gain, has been increasingly used off-label in the palliative care setting to stimulate appetite and improve weight gain, reduce anxiety and insomnia, and relieve chemotherapy-induced nausea and vomiting (CINV). However, adverse effects (AEs) such as drowsiness, hypotension, neutropenia, and increased restlessness have been reported [116].

#### 4.1.2. Steroids

For patients with advanced cancer who cannot tolerate low-dose olanzapine, ASCO also recommends a short-term trial of a progesterone analog or a corticosteroid [5]. Progesterone analogs such as Megestrol Acetate (MA) improve appetite and body weight, but weight gain is primarily due to gain in fat mass rather than skeletal muscle. Furthermore, treatment-related AEs such as thrombo-embolic events, edema, and adrenal suppression limit their use. Corticosteroids have shown benefits in terms of preserving QoL and improving appetite but have not been shown to increase body weight so their use is also limited to end-of-life care owing to the high rate of secondary toxicities, including myopathy, infections, and declined efficacy with prolonged use [117].

#### 4.1.3. Cytokine Modulators

Chronic systemic inflammation is believed to play a prominent role in fat and muscle wasting and cachexia pathogenesis, and increases in levels of tumor-derived pro-inflammatory cytokines such as TNF-α and IL-6 in CAC are well documented [113,114]. Cytokine-blocking antibodies that target TNF-α and IL-6 (i.e., Infliximab and Clazakizumab, respectively) have been assessed in CAC patients. Infliximab trials were terminated due to lack of efficacy, whereas Clazakizumab (previously ALD518) attenuated loss of lean body mass and reversed fatigue in CAC patients; however, no significant differences in OS were observed [116,118]. Several clinical trials assessing the efficacy of the anti-cancer drug thalidomide (a suppressor of TNFα and IL-6 synthesis that possesses anti-inflammatory, immune-modulatory properties) in attenuating weight loss have also been conducted [115,119], with mixed results. Some studies showed attenuation of loss of weight and lean body mass, whereas others showed no differences compared to placebo. Unfortunately, the contrasting results and poor tolerability reported in some trials have so far prevented thalidomide from becoming a viable therapeutic option [117,118]. However, thalidomide in combination with Megestrol Acetate (MA) therapy showed significantly increased body weight, QoL, appetite, and handgrip strength, in cachexia patients compared with MA alone, such combination therapy warrants further investigation.

Another cytokine of interest is GDF-15 (Growth differentiation factor-15; a member of the transforming growth factor-β (TGF-β) cytokine family). High serum levels of GDF-15 represent a poor prognostic factor for cancer patients as circulating levels of GDF-15 have been shown to positively correlate with weight loss, and negatively correlate with lean body and muscle mass. GDF-15 appears to mediate its action through binding to GFRAL (growth factor receptor α-like) in a region of the brain involved in appetite control [120]. Anti-GDF-15 monoclonal antibodies such as Ponsegromab are currently in clinical development [121] to assess their ability to restore appetite and promote weight gain.

#### 4.1.4. Ghrelin and Ghrelin Receptor Agonists

Ghrelin, the gut-derived ‘hunger’ hormone, and its impact on stimulating appetite, has spurred the development of ghrelin mimetics to counteract appetite and weight loss in CAC patients. Anamorelin is a synthetic ghrelin receptor agonist [121] that was recently approved in Japan for the treatment of CAC as it demonstrated efficacy in improving appetite, body weight (gains in lean body mass and fat mass), and QoL in cancer patients in multiple clinical trials [122,123]. However, it has yet to achieve FDA approval because it failed to improve overall survival and muscle function, as shown by the handgrip test and 6 min walk test, and some toxicities including hyperglycemia have been reported [124,125].

#### 4.1.5. Anabolic/Catabolic Transforming Agents (ACTAs)

ACTAs are a new class of pharmacologic agents that include anabolic/catabolic transforming agents such as Espindolol (S-pindolol; formerly known as MT-102). Owing to its intrinsic polypharmacology, Espindolol exhibits a trifecta of beneficial effects in preclinical models of cachexia [126]. Firstly, it attenuates muscle loss via non-selective β-blockade; Secondly, it increases anabolism and muscle growth via partial β2-agonism; and thirdly, it improves fatigue and increases appetite via central 5-HT1A antagonism. The ability of ACTAs like Espindilol to act on multiple measures of cachexia via multiple pathways, positions the broader class of ACTAs as unique among cancer cachexia pharmacologic interventions currently in development [127].

#### 4.1.6. Other

The above list is by no means exhaustive, and other pharmacological agents that target cannabinoid 1 receptor (CB1) [128] or melanocortin 4 receptor (MC4R) [121,122], and non-steroidal anti-inflammatory drugs (NSAIDs) [129], prokinetic/gut motility enhancing agents such as metoclopramide [130], and myostatin blockers [123] are also being investigated for the treatment of cachexia.

### 4.2. Animal Models of Cancer Cachexia

The development of preclinical models exhibiting phenotypes that correlate with clinical features of cancer cachexia has proven to be challenging. Finding a temporal balance between the animal developing malignant tumors and induction of cachectic symptoms, whereby those tumors do not present unethical tumor burdens or result in the rapid death of the subject, has proven to be the greatest challenge [124,125]. Typically, as shown in work from the laboratories of Dr. Zimmers (OHSU) [126] and Dr. Judge (UF) [111,127,128,129,130,131], these models utilize either an allograft/xenograft model to introduce cancer cells into animals or genetically engineer the rodents to develop tumors [124,125].

#### 4.2.1. Allograft Models

Allograft models for CAC are facilitated through the inoculation of cancer cells into rodents, which develop consistent, fast-growing tumors and induce cachectic symptoms [124,125]. One common allograft model is the Lewis Lung Cancer model [132,133]. Following inoculation, the cohort displayed significant loss of body mass within two weeks [132,133,134]. Additional cachectic markers, including degraded and reduced mitochondria, significant upregulation of reactive oxygen species (ROS), and a significant reduction in muscle cross-sectional area were found [132,133,134]. A strategic disadvantage of the LLC cachexia model is the speed at which the tumors grow, with inoculation rapidly resulting in large tumors, with the anorexia nervosa phenotype appearing only in the late stages, immediately prior to death [135], similar to human disease progression.

As with the LLC model, the colon-26 (C26) colorectal cancer model results in a rapid cachectic state [136]. This model results in increased body fat consumption [137] roughly three weeks post-inoculation, similar to that found in humans [127], and has the benefit of being well characterized and optimized [126].

Another model used in CAC research is the Walker 256 rat cancer model (carcinoma) [138]. The Walker 256 tumor has been maintained in vivo for decades and has been routinely used as a primary screen for anti-cancer agents. Following subcutaneous inoculation, symptoms commonly attributed to cachexia such as decreased body weight, reduction in skeletal muscle and adipose tissue mass, decrease in food intake, and a systemic inflammatory response were observed [139]. While effective in producing symptoms common to cachexia, a drawback of this model is the high tumor burden [135].

Another commonly used allograft model is the Yoshida AH-130 model, which induces reproducible, rapid, and progressive tissue wasting. Intraperitoneal inoculation of Yoshida AH-130 ascites hepatoma cells in Wistar rats results in severe anorexia and wasting, with up to 30% weight loss after two weeks of tumor growth [140,141,142,143]. The reproducibility and rapid onset of cachexia [140] allowed for the development of a small molecule agent MT-102, a beta-adrenergic receptor agonist that was found to reduce catabolism and increase anabolism in skeletal muscle, improving survival [140]. Unfortunately, the Yoshida AH-130 model can present with a large amount of ascites, and it has been suggested that the growing ascites decreases food intake by mechanical compression of the gut and, thus, is not a suitable model for investigating CAC [140].

#### 4.2.2. Genetically Engineered Mouse Models (GEMM)

Although the advantages of GEMM are that they usually have a similar genetic background and intact immune system, one drawback of most GEMM for CAC research is that not all mice may develop CAC. Nevertheless, a GEMM commonly used to study CAC is the K-ras^LSL.G12D/+^; Trp53^R172H/+^; Pdx-1-Cre (KPC, KRAS, and p53 double mutant under a pancreas-specific Cre driver) [144,145] KPC mice have been instrumental in elucidating sex differences in cachexia onset and treatment response [58,146] and has also been used to identify markers of cachexia [147]. Furthermore, a cell line derived from a liver metastasis in a KPC mouse (KPCMl1 cells) was used to optimize an orthotopic PDAC model for both high rates of metastasis and CC [148].

A lung cancer GEMM: *Kras^G12D/+^; Lkb1^f/f^* (KL, KRAS mutant and Lkb1^−/−^) [149,150], which demonstrates CC in roughly 60% of rodents, is also used [149]. Sex differences in CAC were also noted in the KL model, suggesting that this phenomenon is not GI cancer-specific [151].

More recently, the *Ptf1a*^+*/ER−Cre*^; *LSL-Kras*^+*/G12D*^; *Pten^f/f^* (KPP, KRAS mutant and PTEN^−/−^ under a pancreas-specific Cre) model was developed as an alternative to the KPC model [152]. Cachectic mice with this genotype recapitulate the muscle gene expression molecular fingerprint of cachectic PDAC patients, unlike allograft models or the KPC model [58,145,152]. Although the KPP model is quite recent, it has been used successfully to study PDAC growth in a nutrient-deprived environment [153].

### 4.3. Nutrition

No curative treatments are available for cancer cachexia in the United States, but nutritional intervention is at the cornerstone of multimodal therapy to counteract muscle mass loss and underlying metabolic pathways of cachexia [154]. To our knowledge, only four dietary intervention studies have been conducted to prevent CAC. Two studies conducted in a variety of cancer populations including lung, breast, and head and neck utilized nutrition coaching and body weight maintenance, and two head and neck cancer studies encouraged the use of nutrition supplements. The findings are mixed. One noted positive associations between nutrition coaching and body weight maintenance in *n* = 10 patients with CAC during treatment [155], while a study conducted in *n* = 58 undernourished cancer patients predominately lung, head and neck, colorectal, and breast cancer was not associated with improvements in body weight [156].

Two studies encouraged the use of nutritional supplements enriched with omega-3 fatty acids [157,158]. In a study of 78 head and neck cancer patients with cachexia, patients were encouraged to consume an oral nutrition supplement enriched with omega-3 fatty acids, micronutrients, and probiotics. While the intervention group consumed fewer daily calories, they were able to significantly increase their body weight as compared to the control population [157]. In another nutrition supplement study, postsurgical head and neck cancer patients (*n* = 19) with recent weight loss >5% were asked to consume three cans of an omega-3 fatty-acid- and arginine-enhanced supplement for twelve weeks [158]. The three-can group experienced improvements in weight, fat mass, and fat-free mass with the supplement use; however, the group randomized to only consume two cans per day for twelve weeks noted no improvements to body composition [158]. Future studies are warranted to determine the appropriate dosage, timing, efficacy, and safety of nutritional interventions for undernourished and cachexic patients.

### 4.4. Exercise and Physical Activity

Exercise interventions have emerged as potential components of comprehensive strategies to prevent and manage CAC. Research among cancer survivors indicates that structured exercise programming can mitigate skeletal muscle wasting by promoting muscle protein synthesis, regulating metabolism, reducing inflammation, and improving physical function [159]. Exercise hypothetically addresses the physiological aspects of CAC while eliciting demonstrable benefits in other tangential aspects of cancer treatment and survivorship, including improving QoL, managing symptoms such as fatigue, and potentially improving treatment tolerance [160]. Insufficient evidence exists to recommend exercise alone for reversing the complex and multifactorial nature of CAC, as highlighted by the 2020 ASCO guidelines for managing cancer cachexia [4], and evidence of benefits from exercise as a component of multimodal interventions to treat CAC is limited [160]. However, there is continued interest in examining the integration of exercise into comprehensive care plans to ameliorate the detrimental effects of cancer cachexia on overall well-being. Cancer patients with good performance status who are capable of physical activity should be encouraged to perform exercises as recommended herein. Furthermore, as ongoing research continues to highlight their potential benefits in the context of CAC, the incorporation of appropriate, tailored, and safe exercise interventions in the standard care paradigm for cancer cachexia remains an important area of research and clinical development.

Exercise oncology interventions, including those implemented among patients who are cachectic or have elevated risk for developing cachexia, typically include both aerobic and resistance training components. The American College of Sports Medicine (ACSM) guidelines for exercise among individuals undergoing cancer treatment include both moderate-intensity aerobic exercise (30 min on at least 3 days per week) and resistance training (at least 2 sets of 8–15 repetitions for all major muscle groups using loads that are at least 60% of maximal strength) [161]. As with any context for exercise oncology intervention delivery, it is important that programming is tailored and progresses according to individuals’ fitness levels and health status [161]. With limited evidence from randomized trials regarding the effectiveness and acceptability of exercise interventions for individuals with CAC, it will be important for future studies to identify achievable exercise volumes that elicit benefits.

### 4.5. Lessons from Nature

As CAC is associated with systemic inflammation and oxidative stress, looking to nature for evolutionary adaptations to chronic oxidative stress and accompanying inflammation may provide novel insights into potential therapies for treating CAC. For instance, animals of many taxa undergo activities at certain stages of their life cycles that pose challenges of oxidative stress and the potential for inflammation. Examples include long-distance migratory birds [162] and hibernating mammals [163,164], among others [165], both of which alter their metabolic programs to accommodate the severe metabolic challenges associated with migration or hibernation [166,167,168,169]. These animal taxa have evolved biochemical, physiological, and behavioral strategies that allow them to meet these challenges and maintain redox balance [162,163,165,170,171,172,173]. Some migratory bird species change their diets to boost consumption of mono- and poly-unsaturated fatty acids, and dietary (exogenous) antioxidants, including anthocyanins and other phenolics, prior to migration or during migration at migratory staging areas [174,175,176]. Similarly, some hibernating mammals also increase consumption of polyunsaturated fatty acids before entering hibernation or torpor [176]. An intriguing study suggests that, at least in migratory birds, exercise may be necessary for dietary antioxidants to reach the mitochondria where they are employed to counter the production of reactive oxygen species (ROS) produced during oxidative phosphorylation [177]. These animal studies suggest that a varied intake of exogenous, dietary antioxidants and consumption of mono- and poly-unsaturated fatty acids, coupled with an exercise regime, should be part of any multimodal therapy strategy to counter the progression of CAC. Lessons we may draw from evolutionary adaptations of animals that, owing to their annual cycles of activities, face repeated bouts of increased oxidative stress, reinforce the major goals of both nutritional and medical exercise intervention strategies discussed above. We should continue to look to nature for inspiration for novel interventions.

### 4.6. Opportunities for Advancement: Multimodal Interventions

It is widely accepted that CAC is a complex, multisystem syndrome that requires a multimodal approach to treatment. The combination of multitargeted, multi-nutrient approaches is a particularly promising route for CAC in experimental models, although more research is needed to determine the optimal quantity, timing, and combination of nutrients [151]. Although nutrition therapy is necessary, alone it may be insufficient for improving body composition, promoting additional or completely new treatment options to rebuild muscle [156]. Multimodal interventions targeted at increasing protein anabolism and decreasing catabolism, such as resistance training, combined with nutritional intervention, hold great promise due to their associations with improvement in lean muscle mass and body composition [157].

Furthermore, multimodal approaches that combine nutrition, exercise, and pharmacologic intervention also show promise. For example, a multimodal cachexia intervention referred to as the MENAC trial (Multimodal-Exercise, Nutrition, and Anti-inflammatory medication for Cachexia) involving late-stage lung and pancreatic cancer patients has just completed a Phase III trial (NCT02330926), the outcomes of which were recently reported at ASCO 2024. The MENAC approach consists of non-steroidal anti-inflammatory (NSAIDs) drugs to reduce inflammation, a physical exercise program (resistance and endurance), as well as dietary counseling and oral nutritional supplements, plus standard care versus standard care alone. In the multimodal intervention group, weight stabilized over a six-week period compared to patients who received standard care alone; however, there was no difference observed in physical activity and muscle mass between study groups [178].

In addition to exercise, nutrition, and pharmacologic agents, multimodal interventions should also include psychosocial support [179]. However, the more ‘modes of intervention’ needed to treat and manage this complex disease, the more challenging the clinical trial design and interpretation of outcomes becomes. Other considerations that add to the complexities of CAC clinical trials include the relatively heterogenous patient population and the standard care that these patients will be receiving [180]. It is now well established that the development of CAC can adversely affect anti-cancer drug therapies. For example, the introduction of immune checkpoint inhibitors (ICI) such as Pembrolizumab has significantly improved the outcome in the treatment of advanced non-small cell lung cancer (NSCLC). However, in multiple retrospective studies, it has been found that time to treatment failure (TTF) and overall survival (OS) are significantly shorter in NSCLC patients with cachexia [181,182]. These findings further underscore the need to improve our understanding of the underlying mechanisms that trigger the genesis of the cachectic syndrome. It may be that further investigation of the reduced responsiveness to immunotherapy and/or molecularly targeted drug therapies may reveal the pathophysiology of CAC that can be exploited in the development of multimodal interventions.

## 5. Conclusions

In this review, our CAC working group highlighted three areas of unmet need that are complementary to and/or enhance priority areas recommended by others [8,9]: The importance of integrating PRO measures in clinical practice to improve CAC-related outcomes throughout the cancer journey; the need to identify novel molecular, imaging, and behavior biomarkers and use emerging omics technologies, ML approaches, and mathematical models to accurately predict cachexia onset and progression in an efficient and accurate way while accounting for demographic and socioeconomic factors; and the need to use a multidisciplinary, multimodal holistic approach to develop and test interventions (pharmacologic, nutritional, and exercise-based) to prevent cachexia progression and to improve QoL and survival. To make progress in these areas, our working group recognizes the importance of stakeholders including patients, patient advocates, and caregivers as recommended by Dr. Zimmers and others [8]. It is also clear that investment by government agencies and industry partners will be instrumental in helping fill gaps in cachexia research. We commend groups that have offered opportunities to fund cachexia research such as the Cancer Research UK-NCI Cancer Grand Challenge and the Pfizer Global Cachexia ASPIRE Competitive Grant Program, and we ask for a call to action for other stakeholders to offer funding opportunities for multidisciplinary cachexia research in the priority areas mentioned in this review.

## Figures and Tables

**Figure 1 cancers-16-02364-f001:**
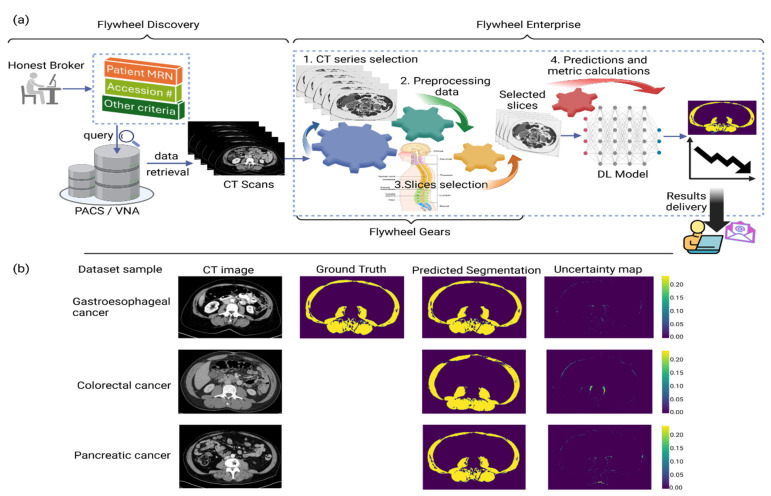
(**a**) Automated pipeline for skeletal muscle segmentation built on Flywheel platform. Processing inside Flywheel Enterprise is undertaken through scripts called “gears”. The data import from PACS/VNA is a manual process. Once the data are imported on Flywheel, the gears start running automatically in a sequence to perform the various tasks: (1) extract the required axial series; (2) data preprocessing; (3) identification of slices corresponding to each vertebra level; (4) slices fed to the trained deep learning model to output the segmented skeletal muscle, along with an uncertainty map that depicts the model confidence. This automated pipeline supports a longitudinal study of patients’ SMI (cm^2^/m^2^) for end users within Moffitt Cancer Center. (**b**) Sample slices at mid-L3 level representative of each dataset of gastroesophageal, colorectal, and pancreatic cancers. In this model, ground truth represents segmentation outputs from the widely validated Sliceomatic Software. The deep learning model output includes the predicted segmentation mask and the uncertainty map depicting the model confidence.

**Figure 2 cancers-16-02364-f002:**
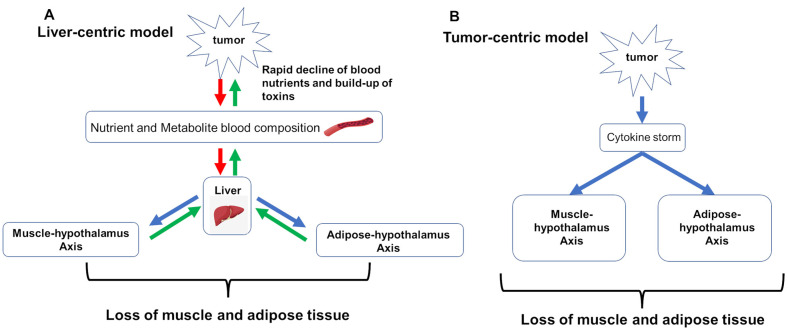
Pathways captured by the mathematical models of cancer cachexia. Red arrows indicate release of toxic or potentially harmful metabolites; green arrows indicate transport or uptake of nutrients; blue arrows indicate signaling. (**A**) hypothesizes that metabolic dysregulation resulting from tumor demand of resources and release of potentially toxic metabolites directly overloads critical functions of the liver. (**B**) tumor-centric model holds that tumor directly induces cancer cachexia via a cytokine storm that drives systemic inflammation which dysregulates muscle and adipose tissue homeostasis, resulting in chronic wasting.

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
