# Peer review of "Defining and Addressing Research Priorities in Cancer Cachexia through Transdisciplinary Collaboration"

_cancers, 2024, doi:10.3390/cancers16132364_

Round 1
Reviewer 1 Report
Comments and Suggestions for Authors
1. In line 41, you have written a comprehensive review of cancer cachexia, and new therapies are needed. However, including one of the new agents being tested in the abstract is inconsistent with the scope of your paper.
2. In line 48, is the primary value of PROs in clinical practice or as an end point for cachexia clinical trials?
3. In line 110, is it practical to use PROs in clinical practice? Are patience compliance and expense considerations in using PROs in routine clinical practice?
4. In line 231, were data for longitudinal weights collected in this study? It would be good to include two reports that describe superior overall survival in advanced non-small cell lung cancer patients who gained or had stable weight while receiving chemotherapy.[Patel J et al. Ann Oncol. & Roeland E et al J Cachex Sarc Musc] Simply measuring longitudinal weights appears to be a reliable indicator of patient outcomes.
5. in line 562, are CRP levels predictive of cachexia, or are they associated with cachexia?
6. In line 516, could differences observed in rates of cachexia be related to biologic differences related to ethnicities, race, and sex, as well as healthcare inequities?
7. In the section starting with line 524, it would good to comment about the MENAC trial that is evaluating exercise, nutritional support, and a non-steroidal anti-inflammatory agent.
8. In line 724, it would be good to include a statement that patients with good performance status and are capable of doing exercise should be encouraged to perform the exercise that you have recommended.
9. In the section that discusses cachexia treatment strategies, it would be good to discuss briefly the complexity of doing cachexia clinical trials and the potentially confounding impact of recent identification of more effective cancer treatments(immunotherapy and genomically targeted therapies).
Author Response
Thank you for taking the time to review our manuscript and provide helpful critiques. Please find detailed responses below. Corresponding revisions/corrections appear via tracked changes in the re-submitted files.
Please note that line numbers mentioned in our responses correspond to the tracked version. Throughout the manuscript we also now abbreviate cancer-associated cachexia as ‘CAC’ rather than ‘CC’ at the request of one of our co-authors.
Reviewer 1
- In line 41, you have written a comprehensive review of cancer cachexia, and new therapies are needed. However, including one of the new agents being tested in the abstract is inconsistent with the scope of your paper.
Response: We thank you for this observation. We have revised the abstract so that it is broader and more in line with the broad scope of our comprehensive review. We no longer reference a particular pharmacologic agent in the abstract to avoid that inconsistency.
- In line 48, is the primary value of PROs in clinical practice or as an end point for cachexia clinical trials?
Response: We have elaborated on the timing and setting of using PROs as an endpoint for cachexia clinical trials. Lines 50-52: The highest priorities identified include the need to (1) evaluate patient reported outcome (PRO) measures obtained in clinical practice and assess their use in improving cancer cachexia-related outcomes.
- In line 110, is it practical to use PROs in clinical practice? Are patience compliance and expense considerations in using PROs in routine clinical practice?
Response: Patient reported outcomes are commonly assessed in clinical practice and include any report of a patient’s health status that comes directly from the patient and can measure symptoms, function, and quality of life. PROs commonly utilized in clinical settings in various ways, including one-time screening questionnaires or to serially monitor a patients’ progress. Touch-screen tablets and internet-based data collection techniques have increased the feasibility of collecting, storing, analyzing, and reporting PROs in real-time in clinical practice. PROs play an important role in the health care system and understanding of health outcomes.
Lines 115-116: (1) Evaluation of patient reported outcome (PRO) measures and examination of their use in predicting cancer cachexia-related outcomes.
Lines 128-133: PROs are data reported directly by patients.9 PRO measures allow for reporting of complex biological and clinical symptoms such as pain, dyspnea, fatigue, and nausea/vomiting, with opportunities for patients to document these concerns routinely via online surveys from the comfort of their own home or in clinic.10 Touch-screen tablets and internet-based data collection techniques have increased the feasibility of collecting, storing, analyzing, and reporting PROs in real-time in clinical practice.
- In line 231, were data for longitudinal weights collected in this study? It would be good to include two reports that describe superior overall survival in advanced non-small cell lung cancer patients who gained or had stable weight while receiving chemotherapy.[Patel J et al. Ann Oncol. & Roeland E et al J Cachex Sarc Musc] Simply measuring longitudinal weights appears to be a reliable indicator of patient outcomes.
Response: Longitudinal weights were not collected in the study. But this is an excellent suggestion for future research that is being planned. Thank you for the suggestion for the inclusion of the two citations which have now been incorporated.
- in line 562, are CRP levels predictive of cachexia, or are they associated with cachexia?
Response: The article that we cite (Ref 95; Mallard et al., 2019) describes CRP as a ‘predictive’ marker based on high levels of CRP being associated with risk of cachexia. On reflection, considering other disorders involving inflammation also exhibit high plasma CRP levels, we agree with the reviewer that it is fair to say that the high CRP levels and cachexia risk findings are more indicative of an association hence we have modified the text to this effect.
- In line 516, could differences observed in rates of cachexia be related to biologic differences related to ethnicities, race, and sex, as well as healthcare inequities?
Response: We agree that such differences could be related to the reasons mentioned and now clarify this important point in line 541-542.
- In the section starting with line 524, it would good to comment about the MENAC trial that is evaluating exercise, nutritional support, and a non-steroidal anti-inflammatory agent.
Response: We thank the reviewer for this suggestion and have added text describing the MENAC trial. We believe a good fit for this text is in section 4.4 which describes multimodal therapies including nutrition and exercise. With the addition of the MENAC trial in the revised manuscript we have moved section 4.4 and created a new section 4.6 to facilitate the flow of the manuscript. This new section begins on line 781.
- In line 724, it would be good to include a statement that patients with good performance status and are capable of doing exercise should be encouraged to perform the exercise that you have recommended.
Response: This is a great point. We have added a statement to this effect on line 739-740 before concluding the paragraph.
- In the section that discusses cachexia treatment strategies, it would be good to discuss briefly the complexity of doing cachexia clinical trials and the potentially confounding impact of recent identification of more effective cancer treatments(immunotherapy and genomically targeted therapies).
Response: Indeed, the design, conduct, and interpretation of cachexia clinical trials is challenging. It seems that confounding issues related to immune checkpoint inhibition are starting to be recognized, and we now comment on these complexities in the new section 4.6 starting on line 804.
Reviewer 2 Report
Comments and Suggestions for Authors
This topical review by Park et al is in my opinion of high interest to readers of Cancers and in particular to clinicians and reseachers in the cancer cachexia (CC) field. The Moffit clinic led consortium performs cutting-edge clinical interdisciplinary research. I enjoyed reading this conceptional review, and readers of Cancerns will like to hear on emerging concepts and research priorities as identified at the consortial meeting, in particular how the authors see the future in the integration of big data coming from onmics, biomarkers, PRO patient data, MRT, metabolocmics, etc.
While I overall like this topical review and status report very much there are two points that require some revisions before publication in my opinion:
1. I understand that the Moffit clinic and researchers from this consortium are leaders in the field. And that they are right to be exited about their work and recent progress. However, I think as a scientific article, a consistent objective style and presentation should be used. At some spots I felt that phrases used were getting too close to and “advertisement” style of the consortium and the Moffit center. I mention a few examples below. I suggest to the authors to go carefully over the text and implement a more neutral/objective style.
2. I understand that this review is also a retreat summary, and that many points were discussed informally and in a brainstorming context. Still, I feel that conclusions presented should be acocmpamied whenever possible by a refernece, or data in a supplement, or at least citing source (unpublished, in preparation, data from xxx). I suggest to the authors to go carefully over the text and give more clearly source/status informations. Again, as for (1), I list a few examples.
With regards to point 1:
Page 2, lines 77-90: “H. Lee Moffitt Cancer Center and Research Institute (Moffitt) is the only National Cancer Institute (NCI)…” I understand lines 77-90 are needed to introduce th Lee Moffitt Cancer Center, and give some background on this review, there is also the risk to present this in a too advertising non-scientific manner. I suggest to carefully check lines 77-90, to delete “only” and instead mentioning other comprehensive Cancer centers. Also, the UFL in Gainesville also has for example a strong cancer chachexia experimental research programme.
Similarly, line 96: “Inspired by reports from these working groups..”. Perhaps a more neutral statement is better, such as “By implementing the feedback given from these working groups…”
-Page 4, lines 155/156 “….service has spearheaded implementation of an integrated supportive care service model within the gastrointestinal (GI) cancer clinic…”
Similarly: You may replace “spearheaded” by a more neutral term, i.e. coordinates, connects…
Examples, regarding Point 2:
-Page 5, lines 222-233 seems to refer to an ongoing (or completed?) pilot study. Important conclusions and results are listed, but no reference given, or the source. I suggest to list at least a PI and publication status (“in preparation?), so that workers in the field can identify the source that gave this conclusion for contacting. Best of course, would be a table with data in this review. In the present form, the listing of p values for conclusions without any other information is not very scientific.
-page 9, lines 422,423 and legend to Figure 2: Can you give here a tentative reference, or at least contact information so that interested readers can email? Similarly, more details on the mathematical models would be helpful.
Author Response
Thank you for taking the time to review our manuscript and provide helpful critiques. Please find detailed responses below. Corresponding revisions/corrections appear via tracked changes in the re-submitted files.
Please note that line numbers mentioned in our responses correspond to the tracked version. Throughout the manuscript we also now abbreviate cancer-associated cachexia as ‘CAC’ rather than ‘CC’ at the request of one of our co-authors.
Reviewer 2
- I understand that the Moffit clinic and researchers from this consortium are leaders in the field. And that they are right to be exited about their work and recent progress. However, I think as a scientific article, a consistent objective style and presentation should be used. At some spots I felt that phrases used were getting too close to and “advertisement” style of the consortium and the Moffit center. I mention a few examples below. I suggest to the authors to go carefully over the text and implement a more neutral/objective style.
Response: Your point is well taken. We very much appreciate the reviewer’s rationale for toning down description of Moffitt. We have revised the text accordingly throughout the manuscript to make it more neutral.
Page 2, lines 77-90: “H. Lee Moffitt Cancer Center and Research Institute (Moffitt) is the only National Cancer Institute (NCI)…” I understand lines 77-90 are needed to introduce th Lee Moffitt Cancer Center, and give some background on this review, there is also the risk to present this in a too advertising non-scientific manner. I suggest to carefully check lines 77-90, to delete “only” and instead mentioning other comprehensive Cancer centers. Also, the UFL in Gainesville also has for example a strong cancer cachexia experimental research programme.
Response: This has been addressed by taking away verbiage in lines 80-84 and incorporating information about our center in a more neutral way in lines 88-89. (Of note, Moffitt is the only NCI-Designated Comprehensive Cancer Center in the state of Florida. We are excited that the University of Florida and the University of Miami/Sylvester Cancer Center have earned NCI Designation, but Comprehensive status has not yet been obtained.)
Similarly, line 96: “Inspired by reports from these working groups..”. Perhaps a more neutral statement is better, such as “By implementing the feedback given from these working groups…”
Response: We apologize for any misunderstanding, but here we are referring to two other cancer centers/cachexia initiatives that inspired us, namely LUNGevity Foundation and University of Rochester. We have modified the text (lines 95-104) to include the institutes names and provide clarity on this point. We also now reference another initiative called Sharing Progress in Cancer Care which also identified cachexia priorities who published after our retreat.
-Page 4, lines 155/156 “….service has spearheaded implementation of an integrated supportive care service model within the gastrointestinal (GI) cancer clinic…”
Similarly: You may replace “spearheaded” by a more neutral term, i.e. coordinates, connects…
Response: As suggested, we have changed ‘spearheaded’ to ‘implemented’.
I understand that this review is also a retreat summary, and that many points were discussed informally and in a brainstorming context. Still, I feel that conclusions presented should be accompanied whenever possible by a reference, or data in a supplement, or at least citing source (unpublished, in preparation, data from xxx). I suggest to the authors to go carefully over the text and give more clearly source/status information. Again, as for (1), I list a few examples.
Response: Thanks for bringing this up. We have revised the manuscript to add more information about sources and their status throughout and specifically for the points mentioned below.
-Page 5, lines 222-233 seems to refer to an ongoing (or completed?) pilot study. Important conclusions and results are listed, but no reference given, or the source. I suggest to list at least a PI and publication status (“in preparation?), so that workers in the field can identify the source that gave this conclusion for contacting. Best of course, would be a table with data in this review. In the present form, the listing of p values for conclusions without any other information is not very scientific.
Response: This pilot study was completed and published in Clinical Cancer Research in 2023. The cited reference is number 38 i.e., Bhatt AS, Schabath MB, Hoogland AI, Jim HSL, Brady-Nicholls R. Patient-Reported Outcomes as Interradiographic Predictors of Response in Non-Small Cell Lung Cancer. Clin Cancer Res. 2023 Aug 15;29(16):3142-3150. doi: 10.1158/1078-0432.CCR-23-0396. PMID: 37233986; PMCID: PMC10425729. We added the reference in line 234 and in line 240 we clarified this was a published study.
-page 9, lines 422,423 and legend to Figure 2: Can you give here a tentative reference, or at least contact information so that interested readers can email? Similarly, more details on the mathematical models would be helpful.
Response: On lines 427-438 we now provide equations to describe the hypotheses and associated mathematical model. Interested readers can contact one of the corresponding authors who will put them in touch with the appropriate co-authors/investigators involved in the mathematical modeling.